# A Brief Review of Recent Results in Arsenic Adsorption Process from Aquatic Environments by Metal-Organic Frameworks: Classification Based on Kinetics, Isotherms and Thermodynamics Behaviors

**DOI:** 10.3390/nano13010060

**Published:** 2022-12-23

**Authors:** Mohsen Samimi, Mozhgan Zakeri, Falah Alobaid, Babak Aghel

**Affiliations:** 1Department of Chemical Engineering, Faculty of Engineering, Kermanshah University of Technology, Kermanshah 6715685420, Iran; 2Department of Chemical Engineering, Faculty of Engineering, University of Sistan and Baluchestan, Zahedan 9816745639, Iran; 3Institut Energiesysteme und Energietechnik, Technische Universität Darmstadt, Otto-Berndt-Straße 2, 64287 Darmstadt, Germany

**Keywords:** metal-organic frameworks (MOFs), arsenic uptake, adsorption kinetics, equilibrium, thermodynamic parameters

## Abstract

In nature, arsenic, a metalloid found in soil, is one of the most dangerous elements that can be combined with heavy metals. Industrial wastewater containing heavy metals is considered one of the most dangerous environmental pollutants, especially for microorganisms and human health. An overabundance of heavy metals primarily leads to disturbances in the fundamental reactions and synthesis of essential macromolecules in living organisms. Among these contaminants, the presence of arsenic in the aquatic environment has always been a global concern. As (V) and As (III) are the two most common oxidation states of inorganic arsenic ions. This research concentrates on the kinetics, isotherms, and thermodynamics of metal-organic frameworks (MOFs), which have been applied for arsenic ions uptake from aqueous solutions. This review provides an overview of the current capabilities and properties of MOFs used for arsenic removal, focusing on its kinetics and isotherms of adsorption, as well as its thermodynamic behavior in water and wastewater.

## 1. Introduction

### 1.1. Prospects for Removing Arsenic Ions from Water

One of the most dangerous elements found in nature is arsenic, a metalloid found in soil. This metalloid can be combined with heavy metals to form sulfur-containing ores [1,2]. Heavy metals are an important and hazardous category of water pollutants for human health [3]. Heavy metals that are commonly found in natural water include cadmium, lead, copper, chromium, mercury, zinc and nickel ions [4,5,6]. In general, the technologies that have been used for the removal of heavy metals are based on physicochemical and biological methods. Whilst, at present, biological methods are considered because of their sustainability and cost-effectiveness [7,8], physicochemical methods such as adsorption, membrane filtration and flotation are still more widely used across the world [9,10,11]. Compared with other technologies used in the removal of hazardous pollutants from water, adsorption technology has attracted more attention due to its easy design, low cost and the recyclability of adsorbents [12,13,14]. In the adsorption process, heavy metal ions are adsorbed on the adsorbent material by physical or chemical bonds. Large specific surface area and high and tunable porosity are two essential properties of an effective adsorbent. Activated carbon is one of the most common adsorbents, which has been used for removing impurities from water for a long time due to its high surface area; however, at present, the use of advanced engineered adsorbents such as carbon nanotubes, polymeric materials and metal oxides have received more attention [15,16,17,18]. Arsenic contamination in water resources, which is mainly the product of industrial activities, is a serious threat to the health of humans and other living beings. A variety of human health problems have also been caused by arsenic contamination, such as diabetes, chronic bronchitis, cardiovascular diseases, peripheral neuropathies, and negative effects on reproduction and hematology [19]. Due to the high mobility and toxicity of arsenic species (As (III) and As (V)), long-term exposure to arsenic-contaminated water may lead to skin cancer, liver damage, and nervous and immunological systems [20,21]. The most common natural sources of arsenic contamination in groundwater are arsenic sulfides, arsenic-rich pyrite, and arsenic-rich iron oxyhydroxide [22]. Considering these risks, the World Health Organization (WHO) announced in 2006 that the maximum value of arsenic in drinking water was 10 μg/L [23,24]. Today, the severe environmental risks of consuming arsenic-contaminated water have led societies to develop effective technologies for removing arsenic ions from aqueous solutions. Adsorption is one of the most efficient methods for removing arsenic from aquatic environments. Adsorption efficiency is also highly dependent on the adsorbent used [25].

### 1.2. Sorbents in Use for Arsenic Adsorptive Removal from the Aquatic Environment

The successful adsorptive removal of arsenic strongly depends on the type of adsorbent materials. Activated carbon (AC) is one of the first conventional adsorbents for the treatment of polluted water [26]. However, there are few studies on the application of this adsorbent for arsenic removal. For instance, in several studies, AC modified with different Fe sources were used for arsenic removal [27,28,29]. In these studies, the Fe_3_O_4_-loaded activated carbon and activated carbons modified with iron hydro (oxide) nanoparticles demonstrated a maximum adsorption capacity of 204.2 mg·g^−1^ and 370 μg·g^−1^, respectively. The results of these studies exhibited a greater increase in the adsorption capacity than unmodified AC. In another work, synthesized magnetized palm shell waste-based AC showed an adsorption capacity of 227.6 mg/g^−1^ for arsenic removal [30]. Recently, some types of modified adsorbents based on zeolites, such as Zr-ZM and FeZr-ZCH, showed that they have high potential for the adsorptive removal of As(V) from aquatic environments [31]. It has been reported that, when using copper exchanged zeolite, the concentration of arsenic remaining in treated water was very low (0.011 mgL^−1^ for As(III) [32]. At present, the use of agricultural and industrial waste materials to remove heavy metals has received much attention due to their cheapness and availability. For instance, an adsorbent prepared from vegetable oil, namely FMSWVOI, exhibited an enhancement in both the uptake of both of the As species after pretreatment with Fe^+2^/H_2_O_2_. The maximum As removal using the FMSWVOI was obtained at Fe^+2^/H_2_O_2_ of 1:17 and 30 min of contact time, with 81% As (III) removal at pH of 2 and 75% As (V) removal at pH of 5 [33]. Hydrated cement, which is commercially available, showed a maximum removal efficiency (>90%) for As (III) at each initial concentration [34]. This adsorbent had a maximum adsorption capacity of 1.92 mg/g for the removal of As from aquatic solutions.

Recently, the use of advanced engineered materials for As removal has received much attention. Carbon nanotubes [35,36,37], metal oxides [38,39], graphene oxide [30,40,41,42] and Metal-organic frameworks (MOFs) have been studied for this purpose. The surface area, and consequently the adsorption capacity, of engineered adsorbents can be modified by adding different organic and inorganic materials to their engineered structure. MOFs, as porous coordination polymers, are a class of advanced engineered materials that are synthesized by combining inorganic materials (metal ions) with organic ligands [43,44]. The unique properties of MOFs, such as their large specific area, tunable pore sizes and adjustable functional groups, have made them a superior class of adsorbents for water treatment applications, and particularly for heavy metal removal [45,46,47,48,49]. MOFs, as advanced engineered adsorbents, are synthesized by combining inorganic materials (metal nodes) with organic linkers. A typical structure of MOFs is shown in Figure 1. 

The unique properties of MOFs and their tenability to achieve the desired characteristics have led to their great potential for various applications. In this review, MOFs, which have been reported in the last decade for the removal of As(III) and As(V), were briefly investigated in terms of their adsorption kinetics, isotherms, and thermodynamic behavior.

## 2. Result and Discussion 

### 2.1. MOFs Used for Arsenic Removal in the Last Decade

In the last decade, several types of MOFs have been reported to be excellent adsorbents for the removal of arsenic ions. These synthesized MOFs, reviewed in this study, are listed in Table 1, in order of publication year. In 2012, Fe−BTC MOF, a MOF based on Fe and 1,3,5-benzenetricarboxylic acid, was synthesized by Zhu et al. [50] using an autogenous pressure synthesis by the solvothermal method. The as-prepared Fe−BTC, with a maximum adsorption capacity of 12.258 mg/g at an optimal pH of 4, was applied as an adsorbent for the removal of arsenic from water. MIL-53(Al), as a new MOF, was synthesized by Li et al. (2014) [51] and used for the adsorptive removal process of As(V) from water. The maximum adsorption capacity of MIL-53(Al) was 105.6 mg/g. The As uptake process, at optimal an pH of 8, reached equilibrium after 11 h. Liu et al. (2015) [52] compared three synthesized-adsorbent ZIFs, including cubic, leaf-shaped and dodecahedral ZIFs, with maximum adsorption capacities of 122.6, 108.5, and 117.05 mg/g, respectively, for As (III) removal efficiency. The adsorption process of As (III) for all synthesized ZIFs types, at a solution pH of 8.5 with an initial arsenic concentration of 80 mg/L, reached equilibrium after 10 h. In 2015, the synthesis of ZIF-8 nanoparticles (varied from 200 to 400 nm) with a high surface area (1063.5 m^2^·g^−1^) was reported by Jian et al. [23] for the adsorptive removal of As(III) and As(V), with maximum adsorption capacities of 49.49 and 60.3 mg/g, respectively. Vu et al. (2015) [53] reported a synthesis of MOF, namely MIL-53(Fe), using HF free-solvothermal methods for As(V) adsorption. The adsorption capacity of MIL-53(Fe) was 21.27 mg/g. The As(V) adsorption process, at a pH of 5, reached equilibrium after 90 min. In another study, MOF-808 nanoparticles (varied from 150 to 200 nm) were synthesized using irradiation with a household microwave and suggested for the adsorptive uptake of As(V) from the solution [20]. The MOF-808 nanoparticles had the maximum arsenic adsorption capacity of 24.83 mg/g at a pH of 4. Another MOF, named UiO-66, was suggested by Wang et al. (2015) [54] for As(V) adsorption. The maximum arsenic adsorption capacity of UiO-66 was 303.4 mg/g at the acidic pH of 2 (Figure 2) [55]. Yang et al. (2017) [56] investigated the fast removal of inorganic arsenic (iAs) from water by suggesting CoFe_2_O_4_@MIL-100(Fe) hybrid magnetic nanoparticles. The maximum capacities of CoFe_2_O_4_@MIL-100(Fe) for the adsorptive removal of As(III) and As(V) were 143.6 and 114.8 mg/g, respectively. The adsorption process of As(III) and As(V) with an initial arsenic concentration of 1 mg/L, reached equilibrium after 12 h. Atallah et al. (2017) [24] presented a synthesis of indium-based MOF, named AUBM-1, and its application for arsenic uptake from water. The maximum adsorption capacity of AUBM-1 at an initial arsenic concentration of 40 mg/L was 103.1 mg/g. The As(V) adsorption process, at a pH of 7, reached equilibrium after 3 h. Huo et al. (2018) [57] reported the fabrication of Fe_3_O_4_@ZIF-8, a core-shell MOF composite, with a high surface area (1133 m^2^ g^−1^) for As(III) removal (with a maximum arsenic adsorption capacity of 100 at an optimal pH of 8) from an aqueous solution. Nasir et al. (2015) [58] presented a synthesized two-dimensional leaf-shaped MOF (ZIF-L nanoparticles) as an inexpensive adsorbent (with adsorption capacity of 43.74 mg/g) for As(III) removal from aqueous solutions. In 2018, Sun et al. [59] presented spherical Fe_2_ Co_1_ MOF-74 nanoparticles (varied between 60 and 80 nm) by the solvothermal method with capacities of 266.52 and 292.29 mg/g for the adsorptive removal of As(III) and As(V) from water, respectively. Continuing this study, after three years, in 2021, the synthesis of a novel composite named δ-MnO_2_@Fe/Co-MOF-74 for As(III) uptake, with a high adsorption capacity of 300.5 mg/g, was reported by Yang et al. [60]. Other MOFs, including SUM-8 [61] (with an adsorption capacity of 152.52 mg/g and an acidic pH of 2), UiO-66(Fe/Zr) [62] (with capacities of 101.73 and 204.1 mg/g for the adsorptive removal of As(III) and As(V), respectively), La-MOF-808 [63] (with adsorption capacity of 217.54 mg/g and pH of 8.32) were also synthesized for the adsorptive removal of arsenic in 2022.

### 2.2. Operational Factors Affecting Adsorption for Arsenic Removal by MOFs

#### 2.2.1. The Effect of Solution pH

The solution pH is a very effective parameter that affects arsenic adsorptive removal capacity. Many studies have been conducted for a better understanding of the effect of this parameter on the adsorption process. For instance, J. Sun et al. [59] studied the adsorption of arsenic species on Fe_2_Co_1_ MOF-74 under various solution pH values. Figure 3a shows the adsorption capacity of As (III) and As(V) on this adsorbent for the pH range (3–10). 

As seen in Figure 3, the As (V) adsorption had a clear decreasing trend with the increase in the pH value, while the As (III) adsorption behavior on this adsorbent was significantly different. These different trends of the adsorption of the arsenic species on Fe_2_Co_1_ MOF-74 at different pH levels are related to the zeta potential changes of the adsorbent surface and the arsenic species present in the solution under different pH values. For As(V), the negative charge of the dominant species of H_2_AsO_4_^−^ in the pH values of 3–7 and HAsO_4_^−2^ and AsO_4_^−3^ in the pH values >7, as well as the decrease in the adsorbent surface potential, led to the decrease in the adsorption amount. As(III) primarily takes its form in neutral HAsO_2_ in the pH < 8 and H_2_AsO_3_^−^ in the pH > 9. Therefore, in pH < 8, the electrostatic attraction has no significant role in the adsorption process. However, a negative charge of H_2_AsO_3_^−^ and a decreasing trend of the zeta potential of the adsorbent surface gradually decreases the adsorption amount in the pH values > 9 [59]. In another work, M. Jian et al. synthesized ZIF-8 nanoparticles and studied the effect of pH on this adsorbent performance for the adsorption of arsenic species. They demonstrated that the pH_IEP_ of synthesized ZIF-8 was around neutral pH of 9.6. The results also showed that the maximum adsorption capacity of the arsenic species on ZIF-8 was at neutral pH values, and considerably reduced as the solution pH increased to alkaline. The decreasing trend of the arsenic adsorption in high pH values was attributed to the negative charge of the adsorbent and, consequently, the electrostatic repulsion between arsenic and ZIF-8 [23,24]. In similar works, the optimum arsenic adsorptive removal capacity on Fe-BTC and MIL-53(Al) MOFs was reported in pH 2–10 and around 8, respectively [50,51].

According to the obtained results, electrostatic attraction is an important and determinative factor during the arsenic adsorption process. Furthermore, considering that the maximum adsorption capacity of arsenic on MOFs is around neutral pH values, it can be concluded that, in most cases, there is no need to adjust the pH in water treatment by MOFs, particularly for the treatment of drinking water, which often has a neutral pH value.

#### 2.2.2. The Effect of Initial Arsenic Concentration

The values of the optimal initial concentration of arsenic in the reviewed studies are summarized in Table 1. J. Li et al. tested the adsorption capacity of MIL-53(Al) for As(V) removal at different initial concentrations of 54, 68, 711, and 2428 μg/L of As(V) solution. The results of this research showed a disparity between the maximum and conditional As removal amount in a practical water purification process. Despite the common concentration of As(V) in natural water (50–200 μg/L), the maximum adsorption capacity of MIL-53(Al) was obtained only for a much higher initial As(V) concentration of 2428 μg/L. Continued testing in low initial concentrations showed that, at a permissible concentration of 10 μg/L (the approval of the WHO for the amount of arsenic allowed in drinking water), the adsorption capacity of MIL-53(Al) was higher than some other adsorbents [51]. In the study of M. Jian et al., arsenic adsorption tests on ZIF-8 nanoparticles were performed at the arsenic initial concentrations of 5 to 100 mg/L and also at low initial concentrations. The results obtained from this work indicated that, at low initial concentrations, the concentration of As(V) was reduced dramatically, from 100 to 2.8 μg/L, by using an adsorbent dose of only o.o6 g/L of ZIF-8. However, such a reduction in the concentration was not achieved for As(III) adsorption, even at higher doses (0.2 g/L) of adsorbent [23]. 

### 2.3. Adsorption Kinetic Studies in Arsenic Removal Using MOFs

The adsorption kinetics of arsenic ions by MOFs have been investigated to identify the adsorption mechanism in the process. Adsorption kinetics display how the rate of dissolved adsorption and the contact time control the arsenic amount at the solution interface. This calculable rate is important for designing the adsorption process. Kinetic models are used to evaluate the data in the study of the adsorption mechanism and the diffusion rate-controlling steps. 

Generally, the mechanism and kinetics of the arsenic uptake process have been studied using several common adsorption kinetic models, including pseudo-first-order (P-F-O), pseudo-second-order (P-F-O), and intra-particle diffusion (I.P-D) mechanism.

The differential equation of the P-F-O kinetic model, which is defined based on solid capacity in the solid/liquid system, can be described as follows [64]:(1)$$\frac{dq_{t}}{dt}=k_{1}\left(q_{e}-q_{t}\right)$$

The linearized form of the above-integral response is represented for the P-F-O kinetic by the following equation:(2)$$\ln\left(q_{e}-q_{t}\right)=\ln q_{e}-k_{1}t$$

The values of $k_{1}$ and $q_{e}$ were calculated using the slope and intercept obtained from plotting $\ln\left(q_{e}-q_{t}\right)$ versus *t*.

In the P-S-O kinetic model, the rate-limiting step is chemisorption, which is dependent on adsorption capacity. The differential equation of the P-S-O kinetic model is as follows [65]:(3)$$\frac{dq_{t}}{dt}=k_{2}{\left(q_{e}-q_{t}\right)}^{2}$$

After integrating the above equation, the P-S-O kinetic model linearly can be presented by the following equation:(4)$$\frac{t}{q_{t}}=\frac{1}{k_{2}q_{e}^{2}}+\frac{t}{q_{e}}$$
where $q_{e}\left(\mathrm{mg}\cdot g^{-1}\right)$ and $q_{t}\left(\mathrm{mg}\cdot g^{-1}\right)$ are the arsenic amount adsorbed onto the MOFs at equilibrium and at various times, $k_{1}\left(\min^{-1}\right)\;$and $k_{2}\left(g\cdot {\mathrm{mg}}^{-1}\cdot \min^{-1}\right)$ are also adsorption rate constant of P-F-O and P-S-O kinetics, respectively.

The values of $q_{e}$ and $k_{2}$ can be calculated using the slope and intercept derived from plotting $\frac{t}{q_{t}}$ versus t. 

The adsorption mechanism of arsenic ions using MOFs can be evaluated by fitting the experimental data on an intraparticle mass transfer diffusion plot, which is suggested by Webber and Morris [66]. 

Based on this mechanism, adsorption occurs in several stages, involving the transfer of solute molecules from the aquatic phase onto the sorbent particles’ surface and then its penetration into the adsorbent. The general steps of absorption are: (1) Sorptive transportation from the solution to the adsorbent surface; (2) Diffusion of the sorptive into liquid films in the solid-solution systems; (3) Diffusion of the sorptive through intraparticle diffusion into the internal pore of the adsorbent; and (4) Adsorption/desorption of the sorptive on/from the surface reaction (surface sites). The I.P-D model is described using the following equation:(5)$$q_{t}=k_{ip}t^{\frac{1}{2}}+C$$
where $t^{\frac{1}{2}}$ is the square root of the time, $q_{t}\left(\mathrm{mg}\cdot g^{-1}\right)$ is the amount of arsenic adsorbed onto the MOFs at time *t*, and $k_{ip}\left(\mathrm{mg}\cdot \min^{-1/2\;}\right)$ is the rate constant of the intraparticle diffusion. The values of $k_{ip}$ and C can be calculated by the slope and intercept obtained from plotting $q_{t}$ versus C.

The accuracy and conformity of the adsorption process to the P-F-O, P-S-O, and I.P-D kinetic models can be checked by the evaluation of the *R^2^* values. All researchers, noted in Table 1, reported that, according to the greater correlation coefficient (*R^2^*) value for P-S-O in comparison with P-F-O, the adsorption rate-controlling step can be the chemical interaction between the functional groups of MOFs and arsenic ions.

### 2.4. Adsorption Isotherm Studies in Arsenic Removal Using MOFs

To describe the distribution of the arsenic molecules at the liquid-solid interface, the adsorption isotherm models of Langmuir (L-I-M), Freundlich (F-I-M), and Temkin (T-I-M) have been investigated. The mentioned isotherm models are given by Equations (6)–(8).
(6)$$\frac{C_{e}}{q_{e}}=\frac{1}{K_{L}q_{max}}+\frac{C_{e}}{q_{max}}$$
(7)$$\log q_{e}=\log K_{F}+\frac{1}{n}\log C_{e}$$
(8)$$q_{e}=\beta_{T}\ln K_{T}+\beta_{T}\ln C_{e}$$
where $q_{max}\left(\mathrm{mg}\cdot g^{-1}\right)$ is one of the Langmuir parameters, indicating the theoretical maximum adsorption monolayer capacity, $q_{e}\left(\mathrm{mg}\cdot g^{-1}\right)$ is the adsorption capacity of the arsenic ions at equilibrium. The other Langmuir constants, $C_{e}\left(\mathrm{mg}\cdot L^{-1}\right)$ and $K_{L}\left(L\cdot {\mathrm{mg}}^{-1}\right)$, are related to the arsenic concentration and affinity of the adsorption sites, respectively. $K_{F}\left(\mathrm{mg}\cdot g^{-1}\right)$ is Freundlich’s constant and the parameter of n in F-I-M, which indicate the intensity of adsorption.

An empirical form of the Freundlich equation is based on multilayer adsorption on heterogeneous surfaces. Different types of isotherms are described by 1/n. These values for arsenic ions were between 0 and 1, indicating favorable adsorption.

The constant parameter, *β_T_* (mg·g^−1^), equals RT/b relevant to the heat of sorption, T(K) is the absolute temperature, b(J·mol^−1^) is the Temkin constant, *R* is the gas universal constant, and *K_T_* (L·g^−1^) is the T-I-M constant. Furthermore, the T-I-M takes into account the analyte-analyte interaction, in which the heat of adsorption decreases linearly with the surface coverage.

The correlation coefficients and R^2^ values can be used to analyze the applicability of the isotherm models to describing the adsorption process. In all of the reports (except As (III) of adsorption using CoFe_2_O_4_@MIL-100(Fe) [56]), the L-I-M fit the experimental data better than the other models, indicating that monolayer adsorption served a vital role in the adsorption of arsenic onto the MOFs.

### 2.5. Adsorption Thermodynamic Studies in Arsenic Removal Using MOFs

Arsenic adsorption onto MOFs is examined at different temperatures to determine whether it is a physisorption or chemisorption process. Equations (9) and (10) can be used to calculate the thermodynamic parameters, including changes in the entropy (ΔS°), standard enthalpy (ΔH°), and Gibbs free energy (ΔG°).
(9)ΔG°=ΔH°−TΔS°
(10)$$\ln K_{c}=\frac{\Delta S{}^{\circ}}{R}-\frac{\Delta H{}^{\circ}}{RT}$$
where $R\;\left(8.314\;J\cdot K^{-1}\cdot {\mathrm{mol}}^{-1}\right)$ and T (K) are the ideal gas constant and studied temperature, respectively. Equation *K_c_* = *q_e_*/*C_e_* is used to express the equilibrium constant (*K_c_*).

All of the thermodynamic parameters, including ΔS°, ΔH°, and ΔG°, can be calculated from the slopes and intercepts of the plot ${\ln\;K}_{c}\;$versus 1/T. All of the reports, summarized in Table 1, (except As (V) adsorption using AUBM-1 MOF [24]) showed a negative value of ∆G° for arsenic adsorption in the studied temperature range, indicating the adsorption process occurs spontaneously and favorably. The decrease in the ΔG° values with the increasing temperature demonstrated that the spontaneous behavior in arsenic adsorption was inversely proportional to temperature. All of the experiments reviewed in this work illustrated that the arsenic adsorption process using MOFs is endothermic, as evidenced by the positive value of ΔH°. In addition, the positive value of ΔS° indicated an increase in the randomness at the solid-solution interface during the adsorption of the analyte onto MOFs.

## 3. Conclusions

Arsenic pollution in water and wastewater, which is mainly the product of industrial activities, is one of the most important threats to the health of humans and other microorganisms. In this review, MOFs, which have been reported in the last decade for the removal of As(III) and As(V), were briefly investigated in terms of their adsorption kinetics, isotherms, and thermodynamic behavior. The study of the adsorption kinetics of the MOFs used for the removal of As from aquatic solutions showed the greater correlation coefficient value for P-S-O in comparison with P-F-O, which demonstrated that the adsorption rate-controlling step can be the chemical interaction between functional groups of MOFs and arsenic ions. In all of the research presented in the past decade, with the exception of CoFe_2_O_4_@MIL-100(Fe), the L-I-M fit the experimental data better than the other models, indicating that monolayer adsorption served a vital role in the adsorption of arsenic onto the MOFs. In addition, the optimal pH value for the removal of As(III)/As(V) from the aquatic environment based on MOFs varied between 2 and 10. At optimal conditions, the maximum adsorption capacity of the MOFs reviewed in this study was also 12.287 to 303.4 mg/g. All of the reports reviewed in the present investigation illustrated that the As adsorption process using MOFs is endothermic, as evidenced by the positive value of ΔH°. In addition, the positive value of ΔS° indicated an increase in the randomness at the solid-solution interface during the adsorption of the analyte onto the MOFs.

## Figures and Tables

**Figure 1 nanomaterials-13-00060-f001:**
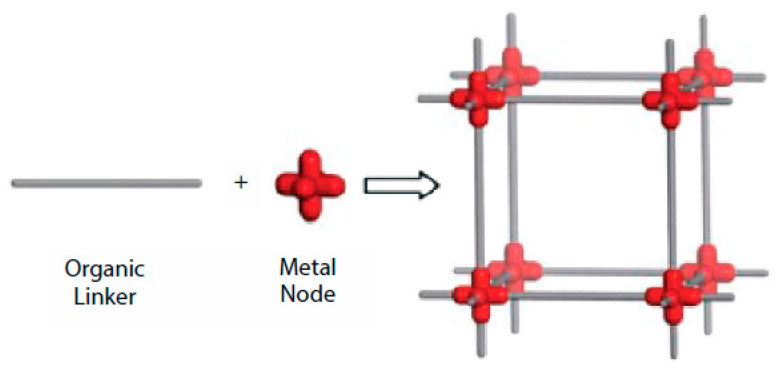
A schematic structure of MOFs. (Reproduced with permission from Ref. [9] with permission from Wiley Online Library, 2022).

**Figure 2 nanomaterials-13-00060-f002:**
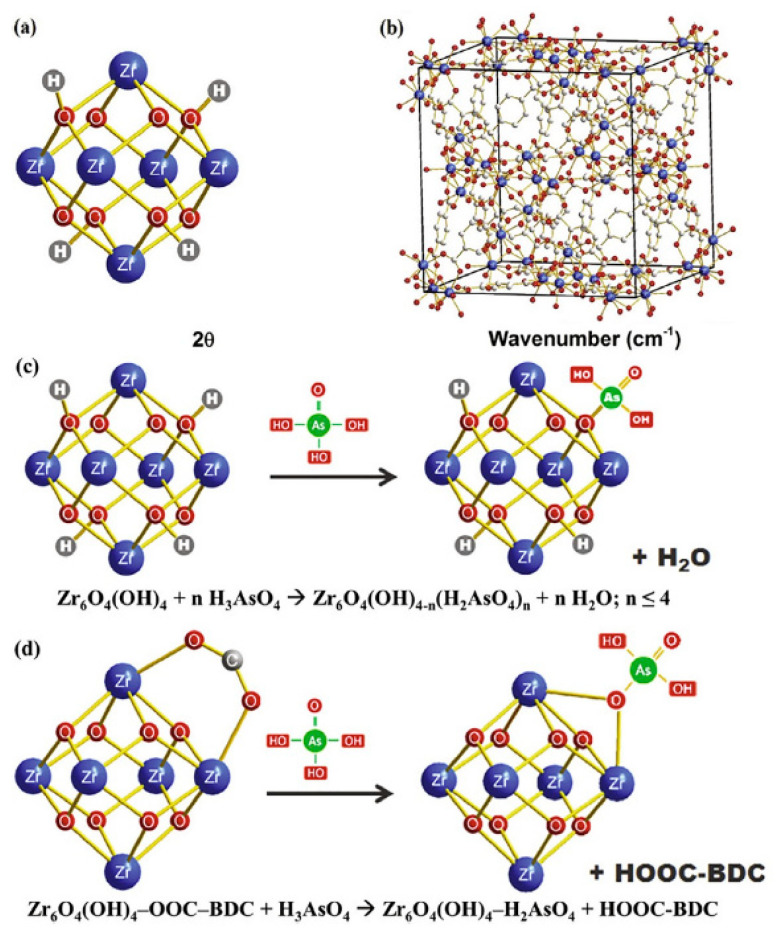
UiO-66 framework (**a**) with octahedral zirconium clusters (**b**), and method of adsorption of arsenate by either coordination at hydroxyl group sites (**c**) or by replacement of BDC ligands (**d**). (Reproduced with permission from Ref. [55] from Elsevier, 2022).

**Figure 3 nanomaterials-13-00060-f003:**
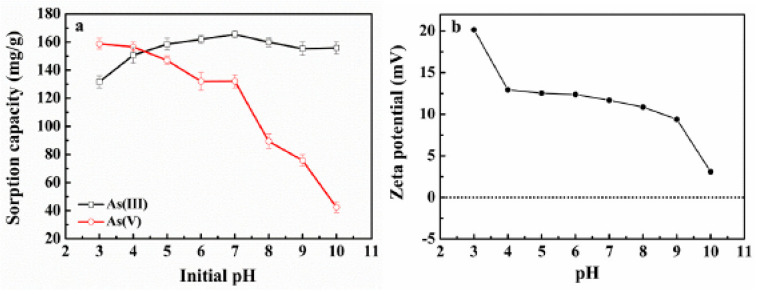
(**a**) The adsorption capacity of As species on Fe_2_Co_1_ MOF-74 under different pH, (**b**) The zeta potential variations of Fe_2_Co_1_ MOF-74 with pH variation. Temperature = 25 °C, adsorbent dose = 0.5 g/L, initial arsenic concentration = 100 mg/L. (Reproduced with permission from Ref. [59] with permission from Elsevier, 2019).

**Table 1 nanomaterials-13-00060-t001:** Comparison MOFs used for arsenic removal from the aquatic environment in the last decade. The Table is sorted based on the year in which the synthesized MOFs were reported.

Adsorbent	Analyte	Optimal Conditions			Adsorption Capacity (mg/g)	Proposed Kinetic Model	Proposed Isotherm Model	Thermodynamic Behavior	Ref.
		pH	Equilibrium Time (min)	Initial Arsenic Concentration (mg/L)					
Fe−BTC MOF	As(V)	4	10	5	12.287	P-S-O	L-I-M	Endothermic process ΔH>0 Spontaneous ΔG<0	[50]
MIL-53(Al)	As(V)	8	660	2.428	105.6	P-S-O	L-I-M	-	[51]
Leaf-shaped ZIFs	As(III)	8.5	600	80	108.5	-	L-I-M	-	[52]
Dodecahedral ZIFs					117.05				
Cubic ZIFs					122.6				
ZIF-8 nanoparticles	As(III)	7	780	100	49.49	P-S-O	L-I-M	-	[23]
	As(V)		420		60.3				
MIL-53(Fe)	As(V)	5	90	5	21.27	P-S-O	L-I-M	-	[53]
MOF-808	As(V)	4	30	5	24.83	P-S-O	-	-	[20]
UiO-66	As(V)	2	1440	50	303.4	-	L-I-M	-	[54]
CoFe_2_O_4_@MIL-100(Fe)	As(III)	2–8	720	1	143.6	P-S-O	F-I-M	Endothermic process ΔH>0 Spontaneous ΔG<0	[56]
	As(V)				114.8		L-I-M		
AUBM-1	As(V)	7	180	40	103.1	P-S-O	L-I-M	Endothermic process ΔH>0 non-spontaneous ΔG>0	[24]
Fe_3_O_4_-ZIF-8	As(III)	8	240	3.5–40	100	P-S-O	L-I-M	Endothermic process ΔH>0 Spontaneous ΔG<0	[57]
2D ZIF-L	As(III)	10	600	20–100	43.74	P-S-O	L-I-M	-	[58]
Fe_2_Co_1_ MOF-74	As(III)	4.3	720	1–250	266.52	P-S-O	L-I-M	-	[59]
	As(V)				292.29				
δ-MnO_2_@Fe/Co-MOF-74	As(III)	10	1440	5	300.5	P-S-O	L-I-M	-	[60]
SUM-8	As(V)	2	720	20	152.52	P-S-O	L-I-M	-	[61]
UiO-66(Fe/Zr)	As(III)	7.1	120	30	101.73	P-S-O	L-I-M	-	[62]
	As(V)			50	204.1				
La-MOF-808	As(V)	8.32	720	100	217.54	P-S-O	L-I-M	Endothermic process ΔH>0 Spontaneous ΔG<0	[63]

## Data Availability

All data is contained within the manuscript.

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
