# Peer review of "A Brief Review of Recent Results in Arsenic Adsorption Process from Aquatic Environments by Metal-Organic Frameworks: Classification Based on Kinetics, Isotherms and Thermodynamics Behaviors"

_nanomaterials, 2022, doi:10.3390/nano13010060_

Round 1

Reviewer 1 Report

Paper entitled “A Brief Review of Recent Results in Arsenic Adsorption Process from Aquatic Environments by Metal-Organic Frame works: Classification Based on Kinetics, Isotherms and Thermodynamics behaviors” meets the necessary standards for publication in this journal.

I recommend supplementing the introduction with other materials used to remove arsenic. Some articles for example:  1.     Georgiana Mladin, Mihaela Ciopec, Adina Negrea, Narcis Duteanu, Petru Negrea, Paula Ianasi, Cătălin Ianasi, Silica- Iron Oxide Nanocomposite Enhanced with Porogen Agent Used for Arsenic Removal, Materials, 2022, 15, 5366, pp. 1-23.  2.     Mihaela Ciopec, Gabriela Biliuta, Adina Negrea (autor de corespondenta), Narcis DuÈ›eanu, Sergiu Coseri, Petru Negrea, Makarand Ghangrekar, Testing of chemically activated cellulose fibers as adsorbents for treatment of arsenic contaminated water, Materials, 2021, 14 (3731), pp. 1-19, 2021  3.     Mihaela Ciopec, Corneliu Mircea Davidescu, Adina Negrea, Narcis Duteanu, Gerlinde Rusu, Oana Grad, Petru Negrea, Amberlite XAD7 resin functionalized with crown ether and Fe(III) used for arsenic removal from water, Pure and Applied Chemistry, 2019, 91(3), pp. 375-388  4.     Andreea Gabor, Corneliu Mircea Davidescu, Adina Negrea, Mihaela Ciopec, Lavinia Lupa, Behaviour of silica and florisil as solid supports in the removal process of As(V) from aqueous solutions, Journal of Analytical Methods in Chemistry, 2015, pp. 1-9  5.     Mihaela Ciopec, Adina Negrea, Lavinia Lupa, Corneliu Mircea Davidescu, Petru Negrea, Studies regarding As(V) adsorption from underground watere by Fe-XAD8-DEHPA impregnated resin. Equilibrium sorption and fixed-bed column tests, Molecules, 2014, 19(10), pp. 16082-16101  6.     Mihaela Ciopec, Adina Negrea, Lavinia Lupa, Corneliu Davidescu, Petru Negrea, Paula SfȃrloagÇŽ, Performance evaluation of the Fe-IR-120 (Na)-DEHPA impregnated resin in the removal process of As(V) from aqueous solution, Journal of materials science and engineering, 2011, Vol. 1, nr.4, pp. 421-432  Graphical representations for equilibrium isotherms and for thermodynamic studies are missing.

There is a lot of information about the elimination of arsenic, being a topic of interest.

The authors treated the presented material very superficially. It can be improved.

Please check the entire manuscript carefully for eventual typographical errors.

Attention when writing references. They are not unitary.

Final Conclusion: The paper meets the necessary standards for publication.

Author Response

 Answers to Reviewer#1 comments:

Paper entitled “A Brief Review of Recent Results in Arsenic Adsorption Process from Aquatic Environments by Metal-Organic Frame works: Classification Based on Kinetics, Isotherms and Thermodynamics behaviors” meets the necessary standards for publication in this journal.

The authors would like to thank you for your kindness and positive comment. The authors would like to thank you for carefully reviewing our manuscript and providing us with comments and suggestions to improve the quality of the manuscript. We have modified the manuscript accordingly, and detailed corrections are marked in red on the revised copy.

Comment 1:

I recommend supplementing the introduction with other materials used to remove arsenic. Some articles for example:

1.     Georgiana Mladin, Mihaela Ciopec, Adina Negrea, Narcis Duteanu, Petru Negrea, Paula Ianasi, Cătălin Ianasi, Silica- Iron Oxide Nanocomposite Enhanced with Porogen Agent Used for Arsenic Removal, Materials, 2022, 15, 5366, pp. 1-23.

2.     Mihaela Ciopec, Gabriela Biliuta, Adina Negrea (autor de corespondenta), Narcis DuÈ›eanu, Sergiu Coseri, Petru Negrea, Makarand Ghangrekar, Testing of chemically activated cellulose fibers as adsorbents for treatment of arsenic contaminated water, Materials, 2021, 14 (3731), pp. 1-19, 2021

3.     Mihaela Ciopec, Corneliu Mircea Davidescu, Adina Negrea, Narcis Duteanu, Gerlinde Rusu, Oana Grad, Petru Negrea, Amberlite XAD7 resin functionalized with crown ether and Fe(III) used for arsenic removal from water, Pure and Applied Chemistry, 2019, 91(3), pp. 375-388

4.     Andreea Gabor, Corneliu Mircea Davidescu, Adina Negrea, Mihaela Ciopec, Lavinia Lupa, Behaviour of silica and florisil as solid supports in the removal process of As(V) from aqueous solutions, Journal of Analytical Methods in Chemistry, 2015, pp. 1-9

5.     Mihaela Ciopec, Adina Negrea, Lavinia Lupa, Corneliu Mircea Davidescu, Petru Negrea, Studies regarding As(V) adsorption from underground watere by Fe-XAD8-DEHPA impregnated resin. Equilibrium sorption and fixed-bed column tests, Molecules, 2014, 19(10), pp. 16082-16101

6.     Mihaela Ciopec, Adina Negrea, Lavinia Lupa, Corneliu Davidescu, Petru Negrea, Paula SfȃrloagÇŽ, Performance evaluation of the Fe-IR-120 (Na)-DEHPA impregnated resin in the removal process of As(V) from aqueous solution, Journal of materials science and engineering, 2011, Vol. 1, nr.4, pp. 421-432  Graphical representations for equilibrium isotherms and for thermodynamic studies are missing.

Answer to comment 1:

Thank you for your suggestion. It was done according to this comment. Some related references were used and added to the manuscript.

Comment 2:

There is a lot of information about the elimination of arsenic, being a topic of interest. The authors treated the presented material very superficially. It can be improved.

Answer to comment 2:

Thank you for your suggestion. In this review, MOFs, which have been reported in the last decade for the removal of As(III) and As(V), were briefly investigated in terms of adsorption kinetics, isotherms, as well as thermodynamic behavior. However, it was done according to this comment and as mentioned in the previous answer, the text has been improved and some related references were used and added to the manuscript.

Comment 3:

Please check the entire manuscript carefully for eventual typographical errors.

Answer to comment 3:

Thank you for your suggestion. It was done according to this comment. We did our best to follow your suggestions and for this purpose, the contents were revised and typographical errors were corrected in the revised manuscript.

Comment 4:

Attention when writing references. They are not unitary.

Answer to comment 4:

Thank you for your suggestion. It was done according to this comment. All references were made through the endnote; however, they were rechecked.

Comment 5:

Final Conclusion: The paper meets the necessary standards for publication.

Answer to comment 5:

Thank you for your kindness and positive comment.

Reviewer 2 Report

1.       Arsenic is metalloid!!! Please avoid the use of the  term “heavy metals”

2.       The application of arsenic should be present more detailed

3.       WHO established As concentration as 10 μg/L not  10 mg/L as authors indicated

4.       If its review why authors included the Section materials and methods?

5.       1.2. Materials in use for arsenic adsorptive removal from the aquatic environment a Table need to be added, including the description of sorbents and etc.

6.       Information about MOFs is better to move in the next section

7.       What was the reason to present Fig. 3 which was taken from another study?

8.       Item 2.3. Adsorption studies in arsenic removal using MOFs can be removed from review, is well know information, the references can be given

9.       Did authors of all cited papers used linear form of kinetic and isotherm models? I am not sure if these formulas need to be presented in the manuscript

10.   English need to be checked by native speaker

11.   The novelty of the study should be better emphasized

Author Response

Answers to Reviewer#2 comments:

Comment 1:

1. Arsenic is metalloid!!! Please avoid the use of the  term “heavy metals”

The authors would like to thank you for carefully reviewing our manuscript and providing us with comments and suggestions to improve the quality of the manuscript. We have modified the manuscript accordingly, and detailed corrections are listed below point by point:

Answer to comment 1:

Thank you for your comment. It was corrected according to this comment. The manuscript, especially the abstract and introduction sections, was modified according to this change.

Comment 2:

2. The application of arsenic should be present more detailed

Answer to comment 2:

Thank you for your suggestion. It was done according to this comment. The manuscript, especially the introduction, was modified according to this change.

Comment 3:

3. WHO established As concentration as 10 μg/L not 10 mg/L as authors indicated

Answer to comment 3:

Thank you very much for your comment. It was corrected according to this comment.

Comment 4:

4. If its review why authors included the Section materials and methods?

Answer to comment 4:

Thank you for your comment. The manuscript doesn’t have this section now.

Comment 5:

5. 1.2. Materials in use for arsenic adsorptive removal from the aquatic environment a Table need to be added, including the description of sorbents and etc.

Answer to comment 5:

Thank you for your comment. In this review, MOFs, which have been reported in the last decade for the removal of As(III) and As(V), were briefly investigated in terms of adsorption kinetics, isotherms, as well as thermodynamic behavior. Based on this purpose the synthesized MOFs, reviewed in this study, are listed in Table 1 in order of publication year, including adsorbent, analyte, optimal conditions of adsorption process, maximum adsorption capacity, proposed kinetic and isotherm models, and thermodynamic behavior.

Comment 6:

6. Information about MOFs is better to move in the next section

Answer to comment 6:

Thank you for your suggestion. It was done according to this comment.

Comment 7:

7. What was the reason to present Fig. 3 which was taken from another study?

Answer to comment 7:

Thank you for your comment. In the section of “The effect of solution pH”, various behavior of the Fe2Co1 MOF-74 (presented by J. Sun et al.) in As (V) and As (III) uptake at different solution pH was investigated. According to the Fig. 3b, these different trends of arsenic species adsorption on Fe2Co1 MOF-74 at different pH are related to the zeta potential changes of the adsorbent surface and the arsenic species present in the solution under different pH values.

Comment 8:

8. Item 2.3. Adsorption studies in arsenic removal using MOFs can be removed from review, is well know information, the references can be given

Answer to comment 8:

Thank you for your suggestion. It was done according to this comment and the mentioned section was removed from the manuscript. The next sections also were modified according to this change.

Comment 9:

9. Did authors of all cited papers used linear form of kinetic and isotherm models? I am not sure if these formulas need to be presented in the manuscript.

Answer to comment 9:

Thank you for your comment. In MOFs, which have been reported in the last decade for the removal of As(III) and As(V), presented in Table 1, the pseudo-first-order, pseudo-second-order, intra-particle diffusion (based on Eqs. 1-5), and Langmuir, Freundlich, Temkin (based on Eqs. 6-8) have been used to describe kinetics and isotherm studies of MOFs.

Comment 10:

10. English need to be checked by native speaker.

Answer to comment 10:

Thank you for your suggestion. It was done according to this comment. We did our best to follow your suggestions and for this purpose, the contents were revised and typographical errors were corrected in the revised manuscript.

Comment 11:

11. The novelty of the study should be better emphasized.

Answer to comment 11:

Thank you for your comment. It was done according to this comment. The novelty of the study was mentioned at the end of the introduction section as follows:

“In this review, MOFs, which have been reported in the last decade for the removal of As(III) and As(V), were briefly investigated in terms of adsorption kinetics, isotherms, as well as thermodynamic behavior.”

Reviewer 3 Report

The paper is interesting considering the subject. Some recommendations refers to:

- The Introductions must be extended with some discutions related to: source of arsenic in water/wastewater. The main arsenic pollution sources.

-The point 2.3 are improper discussed considering the article title and the use references from this point (more references are necessary and more data interpretation).

- Extend the conclusions with the most important data interpreted.

Author Response

Answers to Reviewer#3 comments:

The paper is interesting considering the subject. Some recommendations refers to:

The authors would like to thank you for carefully reviewing our manuscript and providing us with comments and suggestions to improve the quality of the manuscript. We have modified the manuscript accordingly, and detailed corrections are listed below point by point:

Comment 1:

The Introductions must be extended with some discussions related to: source of arsenic in water/wastewater. The main arsenic pollution sources.

Answer to comment 1:

Thank you for your comment. It was done according to this comment. The manuscript, especially the introduction, was modified. Based on this comment, the following information was added to the revised manuscript:

“A variety of human health problems have also been caused by arsenic contamination, such as diabetes, chronic bronchitis, cardiovascular diseases, peripheral neuropathies, and negative effects on reproduction and hematology [19]. Due to the high mobility and toxicity of arsenic species (As (III) and As (V)), long-term exposure to arsenic-contaminated water may lead to skin cancer, liver damage, and nervous and immunological systems [20, 21]. The most common natural sources of arsenic contamination in groundwater are arsenic sulfides, arsenic-rich pyrite, and arsenic-rich iron oxyhydroxide [22].”

Comment 2:

The point 2.3 are improper discussed considering the article title and the use references from this point (more references are necessary and more data interpretation).

Answer to comment 2:

Thank you for your suggestion. It was done according to this comment and the mentioned section was removed from the manuscript. The next sections also were modified according to this change.

Comment 3:

Extend the conclusions with the most important data interpreted.

Answer to comment 3:

Thank you for your comment. It was done according to this comment and the following information was added in the conclusions section of the revised manuscript:

“In addition, the optimal pH value for the As(III) / As(V) removal from the aquatic environment based on MOFs used varied from 2 to 10. At optimal conditions, the maximum adsorption capacity of MOFs reviewed in this study was also 12.287 to 303.4 mg/g.”

Round 2

Reviewer 2 Report

The manuscript can be accepted for publication

Reviewer 3 Report

All my recommendations were considered.